# Genome-Wide Identification, Genomic Organization, and Characterization of Potassium Transport-Related Genes in *Cajanus cajan* and Their Role in Abiotic Stress

**DOI:** 10.3390/plants10112238

**Published:** 2021-10-20

**Authors:** Muhammad Hussnain Siddique, Naeem Iqbal Babar, Roshan Zameer, Saima Muzammil, Nazia Nahid, Usman Ijaz, Ashir Masroor, Majid Nadeem, Muhammad Abdul Rehman Rashid, Abeer Hashem, Farrukh Azeem, Elsayed Fathi Abd_Allah

**Affiliations:** 1Department of Bioinformatics and Biotechnology, Government College University Faisalabad, Faisalabad 38000, Pakistan; mhs1049@gmail.com (M.H.S.); naeembabar17@gmail.com (N.I.B.); roshanzameer23@gcuf.edu.pk (R.Z.); Nazianahid@gmail.com (N.N.); usmanijazahmad1246@gmail.com (U.I.); 2Department of Microbiology, Government College University Faisalabad, Faisalabad 38000, Pakistan; saimamuzammil83@gmail.com; 3Sub-Campus Burewala-Vehari, University of Agriculture Faisalabad, Faisalabad 38000, Pakistan; masroorakb@gmail.com; 4Wheat Research Institute, Ayub Agricultural Research Institute, Faisalabad 38000, Pakistan; majidpbg@yahoo.com; 5Botany and Microbiology Department, College of Science, King Saud University, P.O. Box. 2460, Riyadh 11451, Saudi Arabia; habeer@ksu.edu.sa; 6Plant Production Department, College of Food and Agricultural Sciences, King Saud University, P.O. Box. 2460, Riyadh 11451, Saudi Arabia; eabdallah@ksu.edu.sa

**Keywords:** *Cajanus cajan*, potassium transporters, channels, abiotic stress, gene expression, physiochemical analysis

## Abstract

Potassium is the most important and abundant inorganic cation in plants and it can comprise up to 10% of a plant’s dry weight. Plants possess complex systems of transporters and channels for the transport of K^+^ from soil to numerous parts of plants. *Cajanus cajan* is cultivated in different regions of the world as an economical source of carbohydrates, fiber, proteins, and fodder for animals. In the current study, 39 K^+^ transport genes were identified in *C. cajan*, including 25 K^+^ transporters (17 carrier-like K^+^ transporters (KUP/HAK/KTs), 2 high-affinity potassium transporters (HKTs), and 6 K^+^ efflux transporters (KEAs) and 14 K^+^ channels (9 shakers and 5 tandem-pore K^+^ channels (TPKs). Chromosomal mapping indicated that these genes were randomly distributed among 10 chromosomes. A comparative phylogenetic analysis including protein sequences from *Glycine max, Arabidopsis thaliana, Oryza sativa, Medicago truncatula Cicer arietinum,* and *C. cajan* suggested vital conservation of K^+^ transport genes. Gene structure analysis showed that the intron/exon organization of K^+^ transporter and channel genes is highly conserved in a family-specific manner. In the promoter region, many *cis*-regulatory elements were identified related to abiotic stress, suggesting their role in abiotic stress response. Abiotic stresses (salt, heat, and drought) adversely affect chlorophyll, carotenoids contents, and total soluble proteins. Furthermore, the activities of catalase, superoxide, and peroxidase were altered in *C. cajan* leaves under applied stresses. Expression analysis (RNA-seq data and quantitative real-time PCR) revealed that several K^+^ transport genes were expressed in abiotic stress-responsive manners. The present study provides an in-depth understanding of K^+^ transport system genes in *C. cajan* and serves as a basis for further characterization of these genes.

## 1. Introduction

Potassium (K^+^) is an essential inorganic macro-element that may comprise at least 10% of a plant’s total dry weight [1]. It is also abundantly present in the cytosol of cells (i.e., 60–150 mM) [2,3]. It plays a significant role in physiological processes, i.e., anionic group electrical neutralization, osmoregulation, and membrane polarization. The water potential and turgor pressure of a plant are also controlled by the uptake/release of K^+^. Controlling K^+^ transport in phloem vasculature helps to build up an osmotic gradient that results in driving the flow of sugar sap to sink tissues during the photosynthesis of mature leaves. In xylem vasculature, a plant drives the flow of crude sap from roots to shoots by secreting K^+^ into vessels, when there is no transpiration [4]. The role of K^+^ is also important in guard cells’ movement, which regulates stomatal aperture in plants [5]. The storage of K^+^ in the vacuole is important to maintain the concentration of K^+^ in the cytosol by its exchange between these two compartments. However, the optimal concentration of K^+^ must be sustained for the functioning of a cell. In cytoplasm, K^+^ concentration is correlated with a plant’s tolerance to different stresses such as drought [6] and salinity [7]. Additionally, K^+^ concentration is crucial for anion neutralization, which maintains and modulates membrane potential [2]. The accumulation of K^+^ and its movement from soil to other plant parts are followed by the involvement of a complex potassium transport system, which consists of various transporters and channels [8]. Genome-wide identification and characterization studies have revolutionized our understanding of gene families in several plants [8,9,10,11,12,13]. Thirty-five genes previously reported in *A. thaliana* are now considered K^+^ transporting proteins [9,10].

K^+^ channels are multimeric proteins, which are comprised of transmembrane (TM) segments called α- helices; these α-helices are defined on the basis of pore (P) domains. This part of a channel’s conduction pathway is made from the association of multimeric functional proteins with four P domains. A highly conserved motif with the amino acid sequence “GYGD/E” is also present in K^+^ channels. Topologically, fifteen selective channels of K^+^ in *A. thaliana* have been further divided into three subfamilies, including one K^+^ inward rectifier (Kir-like), five tandem-pore K^+^ channels (TPK), and nine voltage-gated shaker channels [9,10,11]. In addition, three sub-families of K^+^ transporters: Trk/HKT family of K^+^ efflux antiporters (KEA) family (six members), KUP/HAK/KT family of K^+^ uptake permeases (thirteen members), and high-affinity K^+^ transporters (one member) are included [9].

Pigeon pea (*Cajanus cajan*) is considered an important legume crop cultivated across different regions, including South America, South Asia, Africa, and Southeast Asia. India and Myanmar are major producers (>80%) of *C. cajan* [12]. It is a good source of proteins, carbohydrates, fat, fiber, essential amino acids, and minerals for humans and animals [13]. Farmers use its green part to feed their domestic animals as fodder [14]. In plants, K^+^ is a vital nutrient that affects the physiological and biochemical processes associated with different stress mechanisms [15,16,17,18,19]. Our understanding (especially in *C. cajan*) of how K^+^ plays a role in abiotic stress tolerance is limited. In *A. thaliana* and *O. sativa*, different channels and transporters of K^+^ have been reported previously. However, there is no information available about the role of K^+^ channels and transporters in *C. cajan* under abiotic stress. The current study was designed to identify and characterize the K^+^ transport system genes in *C. cajan* and to study their possible role in abiotic stress responses.

## 2. Results

### 2.1. Identification of K^+^ Transporters and Channels

In *C. cajan*, after confirming the motifs, thirty-nine genes for K^+^ transport system were identified (Table 1). Further analysis was performed to characterize transporters and channels separately. Out of thirty-nine genes, twenty-five were transporters and the remaining fourteen were channels. All the transporters and channels were renamed according to their respective chromosome number on which their gene sequence belonged, except the shaker family, which was named according to the shaker family of model plant *A. thaliana* (Table 1). Some motifs such as “GYGD”, “TVSS”, and “SPLY” are conserved in a family specific manner, also highlighting some other conserved motifs in K^+^ transporters and channels of *C. cajan* (Figure 1). The logos are based on class-specific motifs and are adopted from a previous report [20]. These motifs represent a consensus sequence from respective protein sequences of *C. cajan*, *A. thaliana*, and *O. sativa.* Therefore, they can be used as identification criteria for these proteins (Materials and Methods). The average molecular weight ranged from 128.0696 to 35.9705 kDa, and the isoelectric point varied from 4.97 to 9.43.

#### 2.1.1. Potassium Transporters

##### KUP/HAK/KT

Different abbreviations are used for this family in different taxa, such as K^+^ uptake permeases (KUPs) in bacteria and K^+^ transporters (KT) and high-affinity K^+^ transporters (HAKs) in fungi [21,22,23]. Seventeen members of this family were identified in *C. cajan*. This number is greater than *A. thaliana* (13) and comparatively smaller than *O. sativa* (25) (Table 1). Protein sequences of KUP/HAK/KT family members were in a range from 706 to 844 amino acids (Table 1). At a genomic level, the number of exons varied from eight to ten. Transmembrane domains (TM) range from 10 to 14 in *A. thaliana* and *O. sativa*, apart from *OsKUP/HAK/KT3*. There are nine to fourteen transmembrane domains in the HAK family of *C. cajan* and an additional k_trans domain (Table 1). Multiple sequence alignment showed the presence of 12 residue stretches (GGTFALYSLLCR) and a conserved motif of K^+^ transporters (GVVYGDLGTSPLY) in the HAK family of *C*. *cajan*.

##### Trk/HKT

Two members of this family were identified in *C. cajan*, named as *CcHKT1* and *CcHKT2*. The HKT family belongs to the Trk superfamily, which has similar topology to K^+^ channels and is mainly involved in the transport of K^+^/Na^+^. Protein sequences of these two proteins were in a range from 507 to 527 amino acids (Table 1). The exon count ranged from one to three. Transmembrane domains (TMs) ranged from nine to ten (Table 1).

##### KEA Family

Six members of the KEA family have been reported in *A. thaliana* [24], named as *AtKEA1* to *AtKEA6*. The KEA family of *C. cajan* also has six members, named *Cc*KEA1 to *Cc*KEA6. The protein sequences of this family ranged from 574 to 1200 amino acids (Table 1). The number of exons varies from nineteen to twenty-one. The transmembrane domain (TM) numbers range from ten to twelve, except KEA5, which has no transmembrane domains.

#### 2.1.2. Potassium Channels

##### Shaker Family

The shaker family was the first to be studied at the molecular level and nine members of this family have been identified in *A. thaliana* [25]. Similarly, nine members of shaker family have been identified in *C. cajan* (Table 1). The length of protein sequences in shaker family members ranges from 622 to 879 amino acids. The number of exons ranges from ten to thirteen. The number of transmembrane domains ranges from four to five. This family is also further classified into four subgroups: SKOR (one member), AKT (four members), KAT (three members), and GORK (one member) (Table 1).

##### TPK and Kir-Like Family

The TPK family is characterized by the presence of a hydrophobic core, containing four transmembrane domains and two P-loops (also known as KCO-2P); however, the Kir-Like family is characterized by a hydrophobic core comprising a 1P-loop and two transmembrane domains [26,27,28]. In *C. cajan*, five members of TPK and Kir-Like were identified (Table 1). The length of protein sequences of this family members ranges from 340 to 423 amino acids. At a genomic level, the exon count ranges from two to three. There are five transmembrane domains. A highly conserved motif (GYGD) is present in all TPK and Kir-Like family members, except for CcTPK1, where phenylalanine (F) is substituted by tyrosine (Y). This motif (GYGD) is necessary for selective K^+^ channels. The occurrence of substitution may affect the function of CcTPK1.

### 2.2. Gene Structure Analysis and Chromosomal Distribution of Potassium Transport-Related Genes

Over long evolutionary time intervals, exon and intron positions are generally conserved in orthologous genes, whereas intron/exon structure is smaller but sufficiently conserved in paralogous genes [29,30,31,32]. The gene structure was analyzed in K^+^ transporters and channels of *C. cajan* to reveal the intron/exon organization and conservation among transporters and channels. Great diversity was observed in exon count (two to twenty-one) and exon length of K^+^ transporters and channels. It was also observed that members of all five families showed similarities in introns (Figure 2). A significant similarity exists between the intron pattern of the AKT groups of the shaker family, while there is no significant difference in exon length among all AKTs. The exons of *CcAKT1.1* and *CcAKT1.2* have almost similar lengths, while the exon lengths of *CcAKT1.3*, *CcAKT2, CcAKT1.1,* and *CcAKT1.2* are different from each other (Figure 2).

In *C. cajan* 26 out of 39 genes were mapped on 10 chromosomes (Figure 3). The positions of thirteen genes (CcHAK13, CcHAK14, CcHAK15, CcHAK16, CcHAK17, CcKEA6, TPK-KCO4, TPK-KCO, CcSKOR, CcAKT1.2, CcKAT1.1, CcAKT1.1, and CcKAT1.2) were not found, as they were present on the scaffold. They are not represented on the map. Chromosome number eight contains a maximum number of genes (five), four transporters, and one channel. Half of the mapped genes are present on chromosomes two, eight, and eleven, while the remaining half genes are located on the remaining seven chromosomes. Chromosomes three, five, and nine contain three genes each, whereas chromosomes four, six, seven, and ten contain one gene each. There is no gene present on chromosome 1 (Figure 3).

*CcTPK-KCO1* and *CcTPK-KCO2* are segmental duplicates of each other, and both are located on chromosome two. *CcHAK3* and *CcHAK5* are tandemly duplicated with each other and are present on chromosome four and chromosome six, respectively. *CcKEA3* and *CcKEA5* are tandem duplicates and are present on chromosome seven and chromosome eleven, respectively. *CcAKT2* and *CcKAT3* are tandem duplicates and are present on chromosome number eight and chromosome number eleven, respectively. *CcHAK11* and *CcHAK12* are segmental duplicates and are present on chromosome eleven. *CcHAK8* and *CcHAK4* are duplicated by tandem duplication, and both genes are located on chromosomes nine and five, respectively (Figure 3).

### 2.3. Comparative Phylogenetic Analysis

Protein sequences of K^+^ channels and transporters of *C. cajan, A. thaliana, G. max, C. arietinum, O. sativa,* and *M. truncatula* were used for phylogenetic analysis (Appendix A). A phylogenetic tree of K^+^ transporters and channels of *C. cajan, A. thaliana, G. max, C. arietinum, M. truncatula,* and *O. sativa* was constructed to find the evolutionary relationship among them using MEGAX through a maximum likelihood method with 1000 bootstrap replicates. Members of shaker family (*CcAKT1.3*, *CcKAT1.1*, *CcKAT1.2*, *CcAKT2*,) showed a relationship with members of same (shaker) family of *G. max* and also shared the same clades. *CcAKT1.2* and *CcKAT3* are orthologous to members of the shaker family of *G. max* but do not share the same clades. *CcGORK* is orthologous to *AtGORK*. *CcSKOR* is orthologous to *AtSKOR* and members of the shaker family of *G. max*. *CcAKT1.1* is orthologous to Medtr4g113530 and *CarAKT1.1*. No members of the shaker family in *C. cajan* are paralogous to each other (Appendix A). However, *CcTPK-KCO2*, *CcTPK-KCO5*, and *CcTPK-KCO3* are orthologous to *G. max* shaker family members and share the same clades, while *CcTPK-KCO4* and *CcTPK-KCO1* are orthologous to *G. max* shaker family members but do not share the same clades. This shows that channels of *C. cajan* have close relationships with their counterparts *G. max* and *M. truncatula* (Appendix A).

In *C. cajan*, members of the HAK family (*CcHAK8*, *CcHAK1*, *CcHAK11*, *CcHAK4*, *CcHAK5*, *CcHAK3*, *CcHAK6*, *CcHAK9*) showed an orthologous relation with *G. max* (Figure 2), while *CcHAK13*, *CcHAK16*, *CcHAK10*, *CcHAK15*, *CcHAK12* all have orthologous relations with *G. max* HAK family members (Appendix A). *CcHAK7* showed an orthologous relation with *Medtr2g008820* and *CcHAK14* with *Medtr6g007697* (Appendix A). *CcHAK2* showed an orthologous relation with CarKUP/HAK/KT5. *CcHAK17* has an orthologous relation with *AtHAK1*. It shows that the HAK family of *C. cajan* has a close relation with its dicot counterparts, i.e., *G. max, M. truncatula*, and *A. thaliana.* No members of the HAK family of *C. cajan* are paralogues to each other (Appendix A). *CcHKT1* is paralogous to *CcHKT2* and shares the same clade (Appendix A).

### 2.4. Promoter Analysis

*Cis*-regulatory elements (CRE) are linear fragments of non-coding DNA. *Cis*-regulatory elements provide the binding sites to transcription factors [33]. They have many localizations, orientations, and activities in relation to genes. Analysis of the promoter region for *cis*-regulatory elements can provide information about the regulatory networks of genes. Promoter regions (1500 bp upstream) of all K^+^ channels and transporters were searched for CREs to understand the regulatory mechanisms. HKT and shaker families are well characterized in model plants, so selected abiotic stress-related *cis*-elements were searched in the promoter region of shaker and HKT family members of *C. cajan* (Table 2).

### 2.5. Physiological Analysis

#### 2.5.1. Chlorophyll Content

Total chlorophyll content was adversely affected by salt stress compared to heat and drought stress. A significant decrease in total chlorophyll content was observed under salt stress. Higher salt stress for prolonged periods results in adverse effects on total chlorophyll content. Salt stress is one of the major factors affecting the overall growth and yield of plants. The higher decrease in chlorophyll content was observed in salt stress (33.82%) followed by drought stress (23.52%) and heat stress (14.7 %) (Figure 4). The reduction in chlorophyll content leads to a lower rate of photosynthesis, halting the growth and development of plants.

#### 2.5.2. Carotenoid Content

Environmental factors are highly detrimental for carotenoid contents, affecting carotenoids in plants. Most of the carotenoid reduction was observed in plants with salt stress followed by drought and heat stress, as compared to control. Carotenoid content in plants with salt stress treatment was reduced by 35.89% followed by 27.62% in drought stress (Figure 4). Salt and drought stresses are the main factors for reducing carotenoid contents in *C. cajan*.

#### 2.5.3. Total Soluble Protein Content

Environmental factors such as salt, drought, and heat cause adverse effects on total soluble protein content. These factors adversely affect total soluble protein content. A significant reduction in total soluble protein content was observed in plants of *C. cajan*, with salt stress showing a reduction of 89.94 %, followed by 62.39 % under heat stress (Figure 4). Salt stress causes damage to protein synthesizing mechanisms and a loss in tertiary structure, leading to the degradation of misfolded proteins. However, this response of plants depends upon the intensity and duration of the stress, plant species, stage of development, and growing season.

#### 2.5.4. Superoxidase Dismutase (SOD), Peroxide (POD), and Catalase Activity (CAT)

Antioxidant activities (POD, SOD, and CAT) were significantly reduced under drought and heat stresses. The antioxidant activity of peroxide was slightly increased under heat stress, while it decreased under drought and salt stresses (Figure 4). The largest POD reduction (44%) was observed in drought stress compared to control. Superoxide dismutase activity was decreased in all three conditions (heat, salt, and drought). The largest reduction was observed in plants treated with heat stress (42.40%), followed by salt (26.63%) and drought stresses (2%). Catalase (CAT) activity was reduced in all three stress treatments, i.e., heat, drought, and salt compared to control. A substantial reduction in catalase activity was observed under heat stress (67.45%), followed by drought (51.73%) and salt stresses (37.13%).

### 2.6. In Silico Expression Analysis

Comparative expression of K^+^ transporting genes in the flower-bud of two different cultivars of *C. cajan* (AKPR303 and AKCMS11) was analyzed. The average for two biological replicates was used for each cultivar. Most of the genes were expressed in both cultivars, except for some genes that showed zero expression. Some HAK family members (HAK-3, -4, -5, -6, -7, -8, -10, -12, -13, -15, -16, -17) and some TPK-KCO family members (KCO-1, -2, -3, and -4) were found to be highly expressed in both cultivars, while other HAK family members, i.e., HAK-9, -11, -14 and KEA family members, i.e., KEA-1, -2, were found slightly expressed in both cultivars. Specifically, *CcHAK15* was highly expressed (up to six units). *CcHAK2*, *CcHKT1*, and *CcKAT3* showed no expression in all three cultivars (Figure 5A).

Various abiotic stresses have adverse effects on plant growth and it has already been reported that K^+^ transporting genes mediate plant response to salt stress [28] and heat stress [29]. Thus, the already-available RNA-seq data were used to analyze the differential expression of potassium-transporting genes under heat stress at two time points (30 min and 3 h), and under salt stress in two cultivars. Two cultivars of *C. cajan* were used, one salt-tolerant cultivar and other salt cultivar. Our results revealed that the potassium-transporting system is involved in abiotic stress response.

In the case of heat stress, most of the genes were downregulated at both time points, except *CcHAK1*, *CcHAK11*, *CcKAT1.1*, and *CcTPK-KCO2*, which were found to be upregulated (Figure 5B).

In the case of salt stress, *CcHAK15* and *CcTPK-KCO3* were highly upregulated in salt-susceptible cultivars and moderately upregulated in salt-tolerant cultivars, while *CcHAK6* was found to be downregulated in salt-tolerant cultivars and upregulated in salt susceptible cultivars. *CcGORK*, *CcHAK1*, *CcHAK9*, and *CcHAK11* were upregulated in salt-tolerant cultivars and downregulated in salt-susceptible cultivars. *CcHAK6*, *CcKEA6*, and *CcSKOR* were downregulated in salt-tolerant cultivars and upregulated in salt-susceptible cultivars. *CcAKT1.1*, *CcKEA3*, *CcHKT2*, *CcHAK2*, *CcHAK4*, and *CcHAK7* were found to be downregulated in both cultivars (Figure 5C). These results indicated the involvement of potassium-transporting genes in the regulation of susceptibility and tolerance of *C. cajan*.

There was no expression change found in *CcKEA1*, *CcKEA2*, *CcKAT1.2*, *CcKAT3*, and *CcAKT1.3* under both heat and salt stress conditions (Figure 5B,C).

### 2.7. Protein–Protein Interaction

Protein–protein interaction was predicted using the STRING database (Figure 6). The associations predicted through this tool are meant to be specific and meaningful, i.e., proteins jointly contribute to a shared function; this does not necessarily mean they are physically bound to each other. For shaker family members, it predicted interactions between inward rectifying-like channel proteins (CcAKT1.1, CcAKT1.3, CcKAT1.1), a weak inward-like channel protein (CcAKT2), and a regulatory subunit (CcKAT3). Similarly, CcKAT3 also interacted with an outward rectifying-like channel protein (CcGORK), which further interacted with an outward rectifying-like channel CcSKOR and inward rectifying-like channel protein CcAKT1.2. Similarly, interactions were predicted among members of KEA, HAK, and TPK/KCO families. In the KEA family, CcKEA2, 4, and 6 form a network, while in the TPK/KCO family, CcTPK/KCO 3 and 5 form a network. In the HAK family, there were two networks, one for CcHAK1,5,6,7,812,13, 15 and another for CcHAK2,9,16. No interaction was predicted for CcKAT1.2, KEA1, KEA 3, KEA 5, CcTPK/KCO1, and CcTPK/KCO2. In an overall network, CcHKT1 demonstrated a co-occurrence interaction with CcHAK5.

### 2.8. Gene Expression Analysis

Real-time amplification (qRT-PCR) was performed to estimate the transcript abundance of selective K^+^ genes in leaf tissues of *C. cajan* under applied abiotic stresses, i.e., heat, salt, and drought (Figure 7). Gene expression of K^+^ transporters and channels was regulated by various abiotic stresses, and changes in expression led to stress tolerance [34]. The shaker family has been widely investigated and studied for its functional characterization and function in abiotic stress response [35]. Based upon RNA-seq data, nine genes including *CcAKT1.1, CcKAT3, CcAKT2, CcGORK, CcHKT1, CcHAK4, CcHAK6, CcHAK15*, and *CcHAK16* were selected for qPCR-based quantification in response to heat, salt, and drought stresses (Figure 6). Expression analysis of shaker family members *CcAKT1.1, CcAKT2, CcKAT3, CcGORK*, and *CcHKT1* was performed under different stress conditions. *CcAKT1.1* was differentially regulated under salt, heat, and drought stresses. *CcAKT1.1* was found to be highly upregulated (up to five-fold change) in response to drought stress, while, in terms of salt stress and heat stress, it was found to be downregulated. *CcAKT2* was observed to be upregulated in response to heat, salt, and drought stresses. *CcHKT1* was found to be highly upregulated in response to heat stress (up to seven-fold change), salt stress (up to four-fold change), and drought stress (up to two-fold change). *CcKAT3* was found to be highly upregulated in the salt stress condition, while, in drought stress and heat stress conditions, it was found to be moderately upregulated. *CcGORK* expression was upregulated in drought stress and salt stress conditions, while, in the case of heat stress condition, it was found to be downregulated (Figure 7). GORK expression is associated with closure of stomata and drought tolerance [36]. Among the transports, *CcHAK4* was observed to be downregulated, while *CcHAK16* was found to be upregulated under the applied stresses. The expression of *CcHAK6* and *CcHAK15* was reduced under heat stress while it was upregulated in response to salt and drought stresses.

## 3. Discussion

In plants, K^+^ takes part in different cellular mechanism such as maintenance of pH, neutralization of protein charges, activation of enzyme, and stabilization of the synthesis of protein. It is also involved in the regulation of turgor pressure, cell enlargement, water potential, movement of stomata, and membrane polarization [37]. For the uptake of K^+^ from soil to its distribution across the plant body, there is a complex system for K^+^ transport. This system is comprised of two types of proteins, namely transporters and channels [9,35,38]. In the present study, 39 genes encoding for K^+^ transport system have been identified in *C. cajan*. There were nine shaker family genes identified in *C. cajan*, which is consistent with previously reported results, such as in *Vitis vinifera* (nine genes) [39], *A. thaliana* (nine genes) [11], *O. sativa* (eleven genes) [16], and *C. arietinum* (eight genes) [29]. Both outward rectifying channel genes (*GORK* and *SKOR*) are identified in *C. cajan* (*CcGORK* and *CcSKOR*). The identification of *CcGORK* in *C. cajan* shows its important role in stomatal movement and distribution of K^+^ across various parts of a plant’s body [40]. Shaker family of *C. cajan* and *A. thaliana* has same domain pattern. Channels of *C. cajan* showed a closed homology with dicot plants (*G. max, A. thaliana,* and *M. truncatula*), which is opposite to that of monocots (*O. sativa*). *C. cajan* channels have a closed homology with *G. max* compared to *M. truncatula* and *A. thaliana.* The results revealed that channels of K^+^ in *C. cajan* are conserved functionally as well in different plant species.

*In C. cajan*, five members of the TPK-KCO family have been identified; this is similar to *A. thaliana* (five members). The TPK family has a 4TM/2P topology. In addition to *A. thaliana*, the TPK family is also identified in many other plants such as *O. sativa, N. tabacum*, *H. vulgare*, *S. tuberosum* and *C. arietinum* [28,41,42,43]. It shows the conservation and importance of this family in K^+^ transport. The highly conserved motif “GYGD”, which is a hallmark of channels in K^+^ transport systems, is highly conserved in all TPKs except *CcTPK1*, where a substitution of phenylalanine (F) occurs in place of tyrosine (Y). The substitution that occurred in *CcTPK1* may affect the function of the protein.

In addition to the KUP/HAK/KT family of *A. thaliana*, various studies have reported the role of the KUP/HAK/KT family in different plants such as *L. japonicas* [44], *V. vinifera* [45], *P. nigrum, S. lycopersicum* [46], *M. crystallinum* [47], *H. vulgare* [48], seagrass [49], and *C. arietinum* [29]. In the *C. cajan* genome, seventeen members of the HAK family were characterized showing the same pattern of domain as their orthologs in *G. max.* All of them possess nine to thirteen trans-membranes (TMs) and an additional k_trans domain, which takes part in the transport of K^+^.

Some transporters, i.e., *OsHAK5, OsHAK1*, and *AtHAK5*, are well identified and involved in the uptake of K^+^ in roots [50,51]. Knockout of *OsHK5* showed reduced uptake of K^+^ influx rate by 80% in 0.1 mM K^+^ solution [52]. *AtHAK5* is the only gene that is involved in the uptake of K^+^ below 0.01 mM. [53]. The *CcHAK1* showed homology with *AtHAK5* (55.25% similarity) and *OsHAK5* (57.88% similarity) in a phylogenetic tree. This suggests that the *CcHAK1* might play a role in helping plants grow in K^+^-limiting conditions. Compared to *AtHAK5* and *AtAKT1*, *AtKUP7* is involved in the uptake of K^+^ [54]. *AtHAK7* shares a phylogenetic cluster with *CcHAK7* (88.69% similarity) and *CcHAK8* (97.43% similarity). It shows that these proteins share functional properties with *AtHAK7*.

In *C. cajan*, two members of the HKT family (*CcHKT1* and *CcHKT2*) are identified. HKT belongs to the superfamily of Trk, which has a similar topology to K^+^ channels and is primarily involved in K^+^/Na^+^ transport. Both members of the HKT family share phylogenetic clusters with *OsHAK1.4* and *OsHAK1.5*, which shows their role in Na^+^ transport [55].

Physiochemical analysis of *C. cajan* showed that total chlorophyll, carotenoids, and total soluble proteins content is affected by different abiotic stresses such as salt, heat, and drought. It was observed that salt stress adversely affects total chlorophyll, carotenoids, and total soluble proteins content [45,56]. Higher Na^+^ accumulation in cytosol results in the damage of the photosystem, causing a substantial reduction in pigment content. Heat and drought stresses also negatively affect the total chlorophyll, carotenoids, and total soluble protein content. Salt stress causes damage to protein-synthesizing mechanisms and proteins undergo accelerated hydrolysis due to the increased activity of protease. This may cause a loss in tertiary structure and the degradation of misfolded proteins by proteasome. Under stress conditions, rubisco is the enzyme reduction activity that reduces protein synthesis and accelerates its degradation. Furthermore, it also interrupts the soluble amino acid and protein ratio in plants. However, this response of plants depends upon the stress duration and intensity, growing season, and developmental stage of plants [57].

Antioxidant activity of superoxide dismutase, superoxide, and catalase was analyzed under abiotic conditions (salt, heat, and drought). Catalase content was greatly decreased under all abiotic stresses; in particular, a higher reduction was observed under heat stress. The antioxidant activity of superoxide dismutase was mostly affected by heat stress followed by salt stress; drought stress showed a smaller effect compared to control conditions. Superoxide content was decreased under drought stress followed by salt stress, while superoxide content was increased under heat stress [58]. Peroxidation of lipids is considered as the major molecular mechanism that causes oxidative injuries to cell structures under abiotic stress. It is further enforced by the production of reactive oxygen species (ROS) such as hydrogen peroxide (H_2_O_2_), singlet oxygen (^1^O_2_), hydroxyl radical (•OH), and superoxide anion (O_2_•^−^), all of which reduce membrane fluidity. Additionally, it promotes an exchange between the lipid bilayer of two halves in phospholipids that facilitate electrolyte leakage. In recent decades, hydrogen peroxide (H_2_O_2_) has gained interest among other ROS because it plays an important role in abiotic stress [59,60].

Under various abiotic stresses, gene expression analysis revealed the role of K^+^ channels and transporters in the homeostasis of K^+^. Gene expression analysis of *CcGORK, CcAKT1.1, CcKAT3*, and *CcHKT1* was performed under stress conditions, as described earlier. *CcGORK,* a drought tolerance-inducing protein, was expressed in guard cells and was upregulated under drought, heat, and salt stresses, confirming its role in stomatal closure to prevent water loss. Anion efflux that depolarizes the guard cells activates GORK expression, and the resultant efflux of K^+^ prevents stomatal closure and water loss [38]. *AKT1* is involved in the uptake and homeostasis of K^+^ [61]. *CcAKT1.1* was upregulated under salt and drought stresses, while, in heat stress, it was slightly downregulated. *KAT3* is involved in the influx of K^+^ along with AKT1 [62]. Gene expression of *CcKAT3* was upregulated under salt, heat, and drought stresses. Under drought stress, *CcKAT3* was upregulated compared to heat and salt stresses. HKT1 is a K^+^/Na^+^ cotransporter [63]. The expression of *CcKHT1* under heat stress was highly upregulated. It was also upregulated under salt and drought stresses, thus confirming its role in Na^+^ uptake and helping plants to reduce the effects of stress.

## 4. Materials and Methods

### 4.1. Screening of Potassium Transport System in C. cajan

The protein sequences of already-identified K^+^ transport genes from *A. thaliana* and *O. sativa* were utilized as a query to screen the K^+^ transporting genes in *C. cajan* [10,16,17]. An NCBI-BLASTp program was used to retrieve the protein sequences. Raw data were manually curated for the elimination of false-positive results. Furthermore, a selective K^+^ filter G-Y-G-D was also manually confirmed in the protein sequences and redundant sequences were removed. Subsequently, the remaining sequences were used for further analysis. For this purpose, the identified sequences were searched for following motifs: Four KUP/HAK/KT transporters; (1) G-[S, T]-E-A-[M, I]-[F, V]-D-L-[G, A]-[F, V], (2) G-D-[LV-I]-[S, G]-T-S-P-L-Y, (3) G-[D, V]-G-[V, I]-[L, I]-T-P-A-S, (4) A-[D, N]-N-G-[E, P]-G, Two for HKTs; (1) [S, T, A]-x-[F, Y, V, L, C]-x-[D, N, S]-G, (2) [G, A]-[Y, F]-[G, A]-x-[V, A, I]-G-[L, M, Y, F]-[S, T] and three for channels; (1) [L, M, V]-[T, A, D]-[T, S]-[V, T]-G-Y-G-D (2) [A, S]-[L, M, V]-Y-[W, F]-[S-C-T]-[I, V, M]-T (3) H-[C,F,T,A]-A-[A,G]-C-[F,I]-N-[Y,F]. The variants of all predicted genes were checked manually and only the largest open reading frames (ORF) were selected for further analysis. Subsequently, genes were further analyzed by different databases, i.e., Pfam database (http://pfam.janelia.org/ (accessed on 14 August 2021)), NCBI conserved domain database (http://www.ncbi.nlm.nih.gov/Structure/cdd/wrpsb.cgi (accessed on 14 August 2021)), and SMART database (http://smart.embl-heidelberg.de/ (accessed on 14 August 2021)). Highly conserved domains containing genes were selected for further analysis. All the genomic information, protein sequences, chromosomal locations, exon numbers, and protein lengths of K^+^ transporters and channels were taken from NCBI.

### 4.2. Motif Recognition and Gene Structure Prediction

MEME (http://meme.sdsc.edu/meme/meme.html (accessed on 14 August 2021)) was used to find conserved motifs in protein sequences and default parameters were used for motif analysis. Weblogo (https://weblogo.berkeley.edu/logo.cgi (accessed on 14 August 2021)) was used to represent the conserved motifs. NCBI was used to retrieve the genomic and coding sequence of all identified genes. Gene Structure Display Server (http://gsds.cbi.pku.edu.cn/ (accessed on 14 August 2021)) was used to schematically represent the exon–intron structure of genes. Analysis of gene structure was performed using selected sequences.

### 4.3. Multiple Sequence Alignment, Phylogenetic Analysis, and Cis-Regulatory Elements Prediction

ClustalW was used for multiple sequence alignment. Phylogenetic analysis was performed using MEGAX Neighbor-Joining with 1000 bootstrap methods to construct phylogenetic trees. iTOL [18] (https://itol.embl.de (accessed on 14 August 2021)) was used to visualize and modify the phylogenetic tree. *Cis*-regulatory elements (CREs) were searched from upstream regions (1000–1100 bp) of all characterized genes. The *cis*-regulatory elements in the promoter region were analyzed by the Plant CARE website (http://bioinformatics.psb.ugent.be/webtools/plantcare/html/ (accessed on 14 August 2021)).

### 4.4. Gene Duplication, Evolutionary Analysis, and Chromosomal Mapping in Cajanus cajan

Gene duplication events were determined by DNAsp and the Ka/Ks ratio was calculated to determine duplication events. Divergence time was also calculated using the Ks and Ka values to predict the evolutionary events. Genomic Loci of putative K^+^ channels and transporters were graphically represented by the MapChart program (http://www.biometris.wur.nl/UK/Software/MapChart/download (accessed on 14 August 2021)).

### 4.5. Plant Growth and Stress Imposition

Seeds of domesticated *C. cajan* were obtained from a local market of Faisalabad, Pakistan. Plants were grown in plastic pots filled with peat moss in a growth chamber. Pots were kept under the following growth conditions: 22 °C night/25 °C day temperature, 8 h dark and 16 h light period, and 68% humidity. Plants were grown in four different treatments groups: control, heat stress, drought stress, and salt stress, each with three replicates. After 12 days of germination, plants were challenged with heat, drought, and salt stresses. Hoagland solution (10 mL/day) was applied to provide necessary nutrients for plant growth. Drought stress was imposed by limiting the water supply for 7 days, while control was supplied with ample water. Salt stress was imposed by supplying 100 mM NaCl solution for 8 days from the 13th day of germination [52]. Plants were kept at 45 °C for 8 h for heat stress and immediately harvested after treatment. All the treatments were in triplicate and the average values were used for succeeding analysis. The leaves of all treated and control plant samples were dipped in liquid nitrogen and stored at −80 °C for further analysis.

### 4.6. Physiological Analysis

#### 4.6.1. Chlorophyll Content

Fresh leaves (0.1 g) were taken, ground, and homogenized in 80% methanol for estimation of chlorophyll content. These samples were kept at 4 °C overnight. Subsequently, a spectrophotometer (Shanghai Yoke Instrument Co., Ltd., Shanghai, China) was used to take the absorbance at various wavelengths, i.e., 645 nm and 663 nm. Then, 80% methanol was used as blank to normalize the absorbance value of the solvent. The following formulas were used to measure chlorophyll *a*, *b*, total chlorophyll, and chlorophyll *a/b* ratios:Chlorophyll *a* (mg g^−1^ f.wt) = [12.7 (OD 663) − 2.69 (OD 645)] × V/(1000) × W(1)
Chlorophyll *b* (mg g^−1^ f.wt) = [22.9(OD645) − 4.68(OD663)] × V/(1000) × W(2)
Total chlorophyll = chlorophyll *a* + chlorophyll *b*(3)
Chlorophyll *a/b* ratio = Chlorophyll *a*/chlorophyll *b*(4)

V = volume of the extract (mL)

W = weight of fresh leaf tissue (g)

#### 4.6.2. Carotenoid Contents

For estimation of carotenoid contents, fresh leaves (0.1 g) were ground and homogenized in 80% methanol. Samples were centrifuged for 15 min and the supernatant was used to measure the carotenoid contents at 480 nm. The following formula was used to measure total carotenoid content:C = (1000A_480_ − 1.63Ca − 104.96Cb)/221(5)

A = absorbance

Ca = alpha-carotene

Cb = beta-carotene

#### 4.6.3. Biochemical Studies

The activity of biochemical entities (superoxide dismutase, superoxide, catalase, and total soluble proteins) was determined by following protocols reported in different studies. Superoxide dismutase activity was measured by following [64]. Peroxidase activity was determined according to [65]. Catalase activity was assayed as described by [65]. The total soluble proteins were determined by the following [66].

### 4.7. In Silico Expression Profiling and PPI Analysis of Potassium Transport Genes

The expression pattern of K^+^ transport genes was analyzed in two different cultivars of *C. cajan* (AKPR303, AKCMS11, TTB7). For this purpose, RNA-seq data (SRR6780151, SRR6780152, SRR6780153, SRR6780154, SRR6785590, SRR6785591) were downloaded from NCBI-SRA (https://www.ncbi.nlm.nih.gov/sra (accessed on 14 August 2021)). Moreover, to analyze the expression pattern of potassium-transporting genes in *C. cajan* under heat stress at different time points (30 min, 03 h) and salt stress in ICP7: salt-tolerant and ICP1071: salt susceptible, data (heat: SRR11856663 SRR11856664, SRR11856665, salt: SRR5451688, SRR5451689, SRR5451690, SRR5451691) were downloaded from the NCBI-SRA database. These paired-end clean reads were mapped to the *C. cajan* genome. *C. cajan* genome was downloaded from NCBI and indexes were built using bowtie2. The cufflinks program was used to calculate the expression level of annotated genes in the reference genome using the GFF format of the genome. The normalized FPKM value was used to construct a heatmap using TBtools. Furthermore, these identified potassium transporters and channels were subjected to STRING v11.0 (https://string-db.org/ (accessed on 14 August 2021)) to analyze network of protein–protein interactions (PPIs). The minimum required interaction score was 0.400, corresponding to medium confidence.

### 4.8. RNA Isolation, Reverse Transcriptase PCR, and Quantitative Real-Time PCR (qRT-PCR)

Total RNA was extracted from leaf samples of *C. cajan* by Trizol method and quantified by Thermo Nanodrop 2000 (Thermo Fisher Scientific, Waltham, MA, USA). One mg of RNA was used for the synthesis of cDNA using the First-Strand Synthesis kit. cDNA was stored at −20 °C for further analysis. qRT-PCR was performed for expression analysis of transcripts level using a qRT-PCR detection system (CFX96 Touch™ Real-Time PCR Detection System, Bio-Rad laboratories, Hercules, CA, USA) with iTaq Universal SYBR Green SuperMix. The “Oligo Calculator”, an online tool (http://mcb.berkeley.edu/labs/krantz/tools/oligocalc.html (accessed on 14 August 2021)), was used to design gene-specific primers, which were further verified by the NCBI-primer BLAST program (https://www.ncbi.nlm.nih.gov/tools/primer-blast/ (accessed on 14 August 2021)). The expression analysis was triplicated for each of the genes and the GAPDH gene was considered as the housekeeping gene [67].

## 5. Conclusions

In plants, K^+^ is one of the major nutrients (NPK), yet it has attracted very little attention from farmers and researchers. *C. cajan* is cultivated on soils that have low nutrients. Various studies have reported on growth and development by K^+^ transport systems in plants. Therefore, a comprehensive investigation into K^+^ homeostasis mechanisms is required in *C. cajan* to understand their influence on plant development, growth, and stress response. The present research focused on revealing K^+^-transporting genes in *C. cajan*. In this study, 39 genes of the K^+^ transport system were identified, comprising K^+^ transporters (25 genes) and K^+^ channels (14 genes), in *C. cajan* on the basis of their sequence and structure similarity with *A. thaliana*. Evolutionary conservation of K^+^ transport system genes in monocots/dicots and legumes/non-legumes was also observed by gene structure and phylogenetic analysis. Gene promoter analysis provided significant information about the importance of K^+^ transport-related genes in response to abiotic stress in *C. cajan.* Physio-chemical analysis provided insights about the contrary effects of abiotic stresses on *C. cajans* under different physiological parameters such as chlorophyll, carotenoids, and total soluble protein and antioxidant activities, i.e., SOD, POD, and CAT under heat, salt, and drought. The antioxidant activity of these compounds was increased under applied stress conditions; catalase activity was significantly decreased under all three stresses. This study provides initial insight into the *C. cajan* K^+^ transport system, which is helpful for further studies hoping to explore all-genes function in different biological processes and under different stress conditions.

## Figures and Tables

**Figure 1 plants-10-02238-f001:**
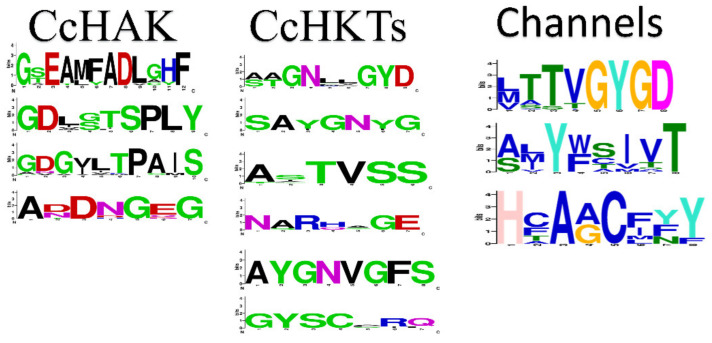
Conserved motif analysis. Sequence logo showing the conserved amino acids of channel and transporter protein sequences from *C. cajan, A. thaliana,* and *O. sativa*.

**Figure 2 plants-10-02238-f002:**
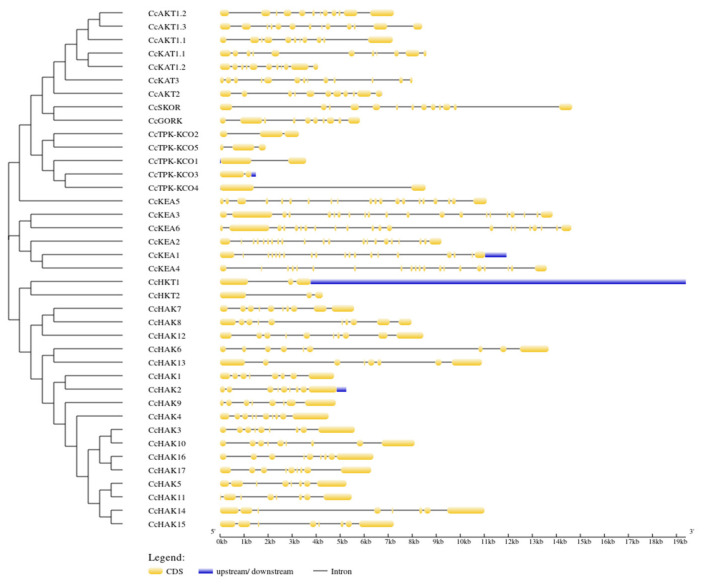
Gene structure of K^+^ transporters and channels of *C. cajan*: yellow boxes show the exons, black lines represent the introns, and blue boxes show untranslated regions (UTRs).

**Figure 3 plants-10-02238-f003:**
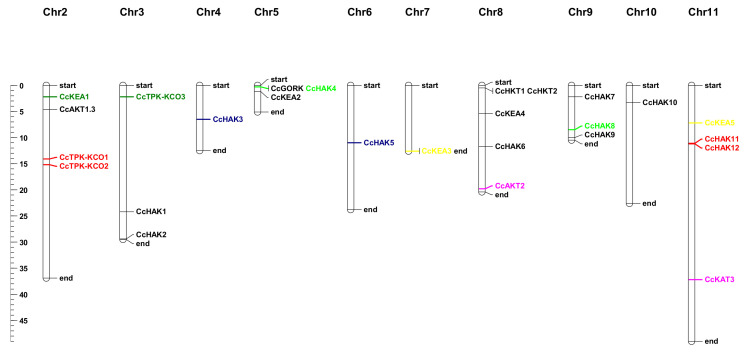
Chromosomal mapping of K^+^ transport system-related genes in *C. cajan* and gene duplication. Segmental duplication is illustrated by red color and tandem duplication is illustrated by multiple colors.

**Figure 4 plants-10-02238-f004:**
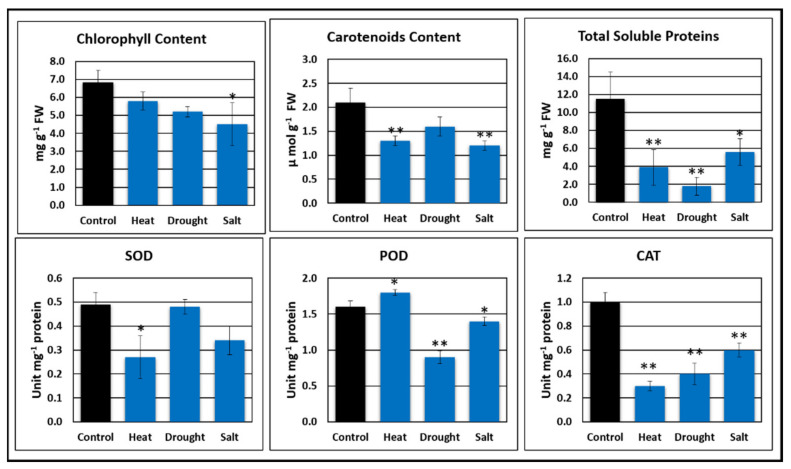
Effect of environmental stresses on different physiochemical parameters of *C. cajan* evaluated by triplicated pot experiments; the average value was used each reading. The standard error is indicated with error bars on each column. * Indicates the significant (*p* < 0.05), while ** indicates the highly significant (*p* < 0.001) differences between environmental stresses and the control for physiochemical parameters.

**Figure 5 plants-10-02238-f005:**
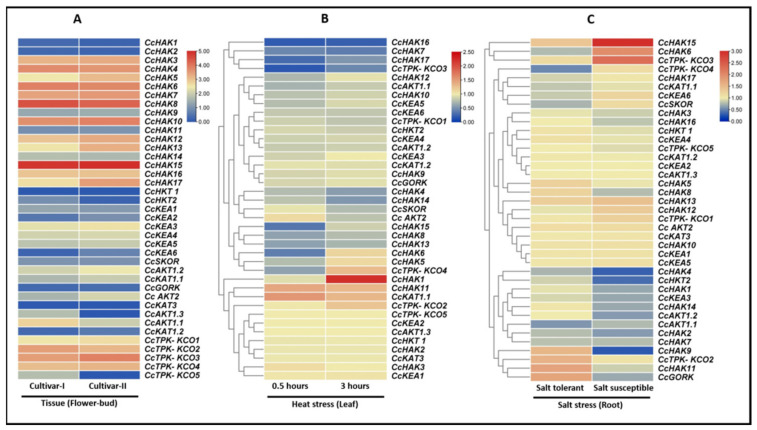
In silico gene expression analysis of potassium transporting genes: (**A**) expression in flower-bud of two cultivars of *C. cajan*, cultivar-I: AKPR303, and cultivar-II: AKCMS11. Red color represents the presence of a gene, while blue color represents an absence of the gene. (**B**) Differential expression from leaf of *C. cajan* against heat stress response at two time points (30 min and 3 h). (**C**) Differential expression from root of salt-tolerant and salt-susceptible cultivars of *C. cajan* against salt stress response. Red color represents the upregulation of genes while blue color represents the downregulation of genes.

**Figure 6 plants-10-02238-f006:**
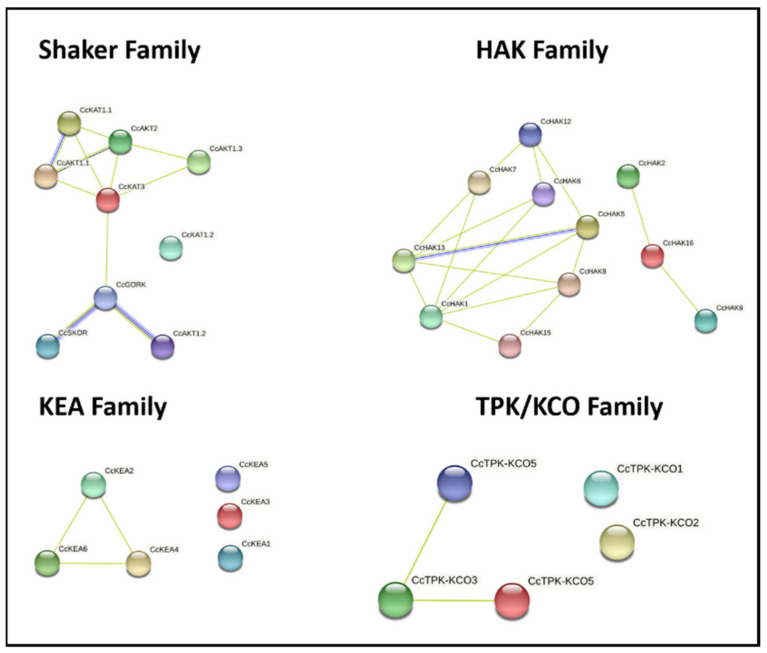
Schematic representation of protein–protein interaction (PPI) networks between potassium transporters and channels from *C. cajan*. Nodes of different colors indicate different proteins.

**Figure 7 plants-10-02238-f007:**
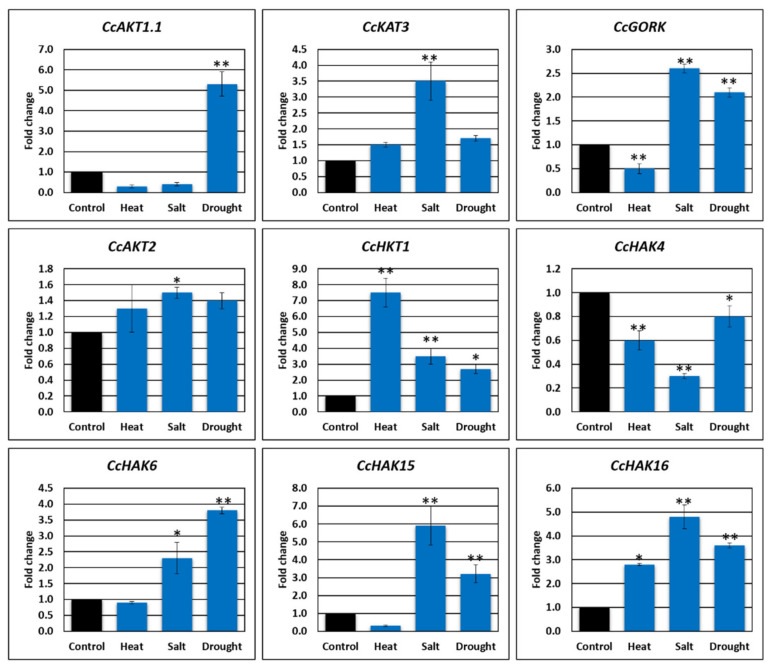
Relative qRT-PCR assay of K+ channel/transporter genes under abiotic (heat, drought, and salt) stresses. The experiment was triplicated to obtain an unbiased average value. The default expression value for each gene was one in non-treated plants. Standard error has been indicated as bars on each column. * Indicates the significant (*p* < 0.05), while ** indicates the highly significant (*p* < 0.001) differences between environmental stresses and the control for physiochemical parameters.

**Table 1 plants-10-02238-t001:** Nomenclature and properties of K^+^ channels and transporters in *C. cajan*.

Sr.#	Gene Name	* Protein ID	TM Domains	Domains	Protein Length	Chromosome Number	Exons	Isoelectric Point	Molecular Weight (kDa)
1	CcHAK1	XP 020213425	12	K_trans	784	3	8	8.56	87.6626
2	CcHAK2	XP 020212862	12	K_trans	804	3	10	8.83	89.3791
3	CcHAK3	XP 020214355	12	PLN00148	784	4	9	8.79	87.4835
4	CcHAK4	XP 029127280	9	K_trans	706	5	10	8.69	78.7490
5	CcHAK5	XP 020218079	13	K_trans	778	6	8	8.45	86.9579
6	CcHAK6	XP 020220430	14	K_trans	792	8	9	7.63	88.4431
7	CcHAK7	XP 020222401	13	PLN00151	841	9	10	6.93	93.4439
8	CcHAK8	XP 020222257	13	PLN00151	844	9	10	5.97	93.9803
9	CcHAK9	XP 020222600	12	K_trans	713	9	8	8.93	79.5610
10	CcHAK10	XP 020223374	10	K_trans	790	10	9	9.30	88.1669
11	CcHAK11	XP 020225584	12	K_trans	773	11	8	8.79	86.3152
12	CcHAK12	XP 020226102	12	PLN00151	841	11	10	8.50	93.5142
13	CcHAK13	XP 029130105	14	K_trans/	790	Unknown	8	8.13	88.5552
14	CcHAK14	XP 020207687	13	PLN00149	773	Unknown	8	7.29	86.9488
15	CcHAK15	XP 020232137	12	PLN00149	781	Unknown	8	7.09	87.4695
16	CcHAK16	XP 020234883	11	K_trans	760	Unknown	9	6.24	84.7841
17	CcHAK17	XP 020236481	12	K_trans	776	Unknown	9	6.89	86.3213
18	CcHKT 1	XP 020220422	10	TrkG	527	8	4	9.35	59.3970
19	CcHKT2	XP 020222038	9	N/A	507	8	3	9.43	57.4761
20	CcKEA1	XP 020227261	12	Na_H_Exchanger	638	2	21	7.23	69.2591
21	CcKEA2	XP 020215520	11	Na_H_Exchanger	586	5	21	5.90	63.0490
22	CcKEA3	XP 020218588	10	SMC_N- PRK03562	1193	7	21	4.97	128.0696
23	CcKEA4	XP 020220606	10	Na_H_Exchanger	574	8	20	6.33	62.3534
24	CcKEA5	XP 020225548	0	PRK03562	815	11	20	5.75	88.8492
25	CcKEA6	XP 020233144	10	PRK03562 -	1200	Unknown	21	4.98	129.4016
26	CcSKOR	XP 020202567	5	ANK-Ank_2	837	Unknown	13	6.68	95.9104
27	CcAKT1.2	XP 020205453	5	ANK-Ank_2	879	Unknown	12	6.02	99.3995
28	CcKAT1.1	XP 020208418	5	Ank_2	717	Unknown	11	7.02	83.3877
29	CcGORK	XP 020215636	5	ANK/Ank_2	808	5	11	8.71	92.3245
30	Cc AKT2	XP 020220877	5	ANK/Ank_2	825	8	11	6.96	94.5856
31	CcKAT3	XP 020225577	5	KHA/Ank_2	622	11	13	7.02	71.5766
32	CcAKT1.3	XP 020227900	5	N/A	864	2	13	6.50	97.5863
33	CcAKT1.1	XP 020231629	5	ANK/Ank_2	878	Unknown	12	6.77	87.5007
34	CcKAT1.2	XP 029125234	4	Ank_2	759	Unknown	11	8.56	99.6863
35	CcTPK- KCO1	XP 020205106	5	Dockerin_like/ Ion_trans_2	359	2	2	8.79	40.0668
36	CcTPK- KCO2	XP 020222684	5	Ion_trans_2-EFh_CREC	348	2	4	5.91	38.8132
37	CcTPK- KCO3	XP 029126956	5	Ion_trans	324	3	2	9.39	35.9705
38	CcTPK- KCO4	XP 020232959	5	Ion_trans	423	Unknown	3	8.72	46.8026
39	CcTPK- KCO5	XP 020230725	5	Ion_trans_2	340	Unknown	3	5.40	38.1138

* Proteins are ordered and grouped according to “Domain” category. Subcellular localization (reliability) was observed in plasma membrane for all proteins.

**Table 2 plants-10-02238-t002:** Promoter analysis of selected K^+^ channels and transporters.

Regulatory Element	Core Sequence	*CcAKT1.1*	*CcAKT1.2*	*CcAKT1.3*	*CcAKT2*	*CcKAT1.1*	*CcKAT1.2*	*CcKAT3*	*CcGORK*	*CcHKT1*	*CcHKT2*	Function
ABRE	CACGTG	1			1			1	1			Response to Abscisic acid signals
	ACGTG	1			2	1		3	3		1	
MYB	TAACCA		2	4	2	4	2	2	2			Response to drought stress and ABA signals
	CAACCA		1	1	1		2	1	2	2	1	
MYC	CATTTG		2	2	2	3	2	3	3	2	3	Response to drought, ABA, and cold signals
W-box	TTGACC			1				1	1			Response to SA, GA, and pathogenesis signals
GT-1 motif	GGTTAA	1	1	1	3	4	1	1	1			Light responsive element
G-box	CACGTG	1			3			3	3	1		Involved in light response
GARE	TCTGTTG							1				Gibberellin-responsive element
MBS	CAACTG					1	2					Involved in drought-inducibility
ARE	AAACCA		1	1	1	1			4	1		Essential for the anaerobic induction
TCA-element	CCATCTTTTT	2	1	1						1		Response to salicylic acid
TC-rich repeats	ATTCTCTAAC		2	2							2	Involved in defense and stress response

## Data Availability

Data is contained within the article and Appendix A.

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
