# Peer review of "Genome-Wide Identification, Genomic Organization, and Characterization of Potassium Transport-Related Genes in Cajanus cajan and Their Role in Abiotic Stress"

_plants, 2021, doi:10.3390/plants10112238_

Round 1
Reviewer 1 Report
This paper addresses potassium transporters and channel proteins in the legume, C. cajan (pigeon pea). The topic is interesting and a comparative approach is taken with models, such as Arabidopsis and rice, that broadens the interest to readers. However, the paper also has several aspects that need to be improved, most notably with respect to describing the expression datasets and in several cases with providing enough information for interpretationof the figures.
Here are the issues:
- Figure 1 does not indicate which logo sequences come from which of the three species. Not clear what is being shown here.
- Figure 2. Does not give the bootstrap support values for the branches of the tree. They may be there, but the font is too small to see. A tree has no real meaning if the reader cannot see the numbers at the base of the named branches. The"Shaker" group currently is not one group. Perhaps it will be if the authors re root the tree, but currently it is just a bunch of basal branches that are outside the other groups.
- Abstract says that the genes are "randomly distributed" along the chromosomes. Am not sure what authors are trying to say with that. Fig. 4 seems to show that the genes are widely spread or evenly distributed, but since so many genes are not mapped to chromosomes, it seems premature to claim a distribution.
- Figs 5 and 7 lack sufficient legends. They do not say sample sizes (how many independent samples measured)? They don't indicate whether the error bars are standard errors or something else, or what bars are significantly different from each other.
- Last, the results section doesn't clearly state what data are generated by the authors and what is derived from the published RNAseq data. An extra sentence or two in places would help to clarify the source will be helpful.
Minor comments:
Commas are not correctly placed in several sentences. A grammar checker out to show those issues. Ex. "Like,....," in the introduction.
For Table 2. Try putting the headers for the genes at a 45 degree angle. Currently hard to read those names.
Reviewer 2 Report
This manuscript explains genome-wide identification, genomic organization, and characterization of potassium transport-related genes in cajanus cajan and their role in abiotic stress. This study identified 39 K+ transport genes in C. cajan, including 25 K+ transporters (17 HAKs, 2 HKTs, 6 KEAs), and 14 K+ channels (9 Shakers and 5 TPKs). Chromosomal mapping indicated that these genes were randomly distributed among 10 chromosomes. A comparative phylogenetic analysis including protein sequences from Glycine max, Arabidopsis thaliana, Oryza sativa, Medicago truncatula Cicer arietinum, and C. cajan suggested strong conservation of K+ transport genes. Gene structure analysis showed that intron/exon organization of K+ transporter and channel genes is highly conserved in a family-specific manner. Many cis-regulatory elements were identified in the promoter region related to abiotic stress, which suggests their role in abiotic stress response. Abiotic stresses (salt, heat, and drought) adversely affected the chlorophyll, carotenoids contents, and total soluble proteins. Furthermore, catalase, superoxide, and peroxidase activities were altered in C. cajan leaves under applied stresses. Expression analysis (RNA-seq data and quantitative real-time PCR) revealed several K+ transport genes were expressed in abiotic stress-responsive manners. The present study gives an in-depth understanding of K+ transport system genes in C. cajan and serves as a basis for further characterization.
The manuscript needs improvement before publication, I would like to add comments to better the manuscript.
- It would be better if the author tries to validate the function of at least one gene they found in this study for its role in abiotic stress.
- Make one hypothetical figure which depicts the findings of this study.
- I suggest doing a putative protein-protein interaction study in-silico.
4. The introduction is short. The author should include recent genome-wide studies such as:
a. Genome-Wide Identification and Characterization of PIN-FORMED(PIN) Gene Family Reveals Role in Developmental and Various Stress Conditions in Triticum aestivum.
b. Genome-wide identification and expression pattern analysis of the KCS gene family in barley.
c. Genome-wide identification and characterization of abiotic stress responsive lncRNAs in Capsicum annuum.
d. Genome-Wide Identification and Characterization of the Brassinazole-resistant (BZR) Gene Family and Its Expression in the Various Developmental Stage and Stress Conditions in Wheat (Triticum aestivum).
e. Genome-wide identification and functional characterization of natural antisense transcripts in Salvia miltiorrhiza.
f. Genome-wide identification and expression analysis of the AT-hook Motif Nuclear Localized gene family in soybean.
Change at
L26 suggested strong conservation to suggested vital conservation.
L30 Abiotic stresses (salt, heat, and drought), adversely to Abiotic stresses (salt, heat, and drought) adversely.
L42 processes i.e., anionic to processes, i.e., the anionic.
L70 different regions including to different regions, including.
L76 plays role in to plays a role in.
L93 isolelectric point to isoelectric point.
L140 transmembrane domains is five to transmembrane domains are five.
L141 members except CcTPK1 to members except for CcTPK1.
L142 K+ channels The occurrence to K+ channels. The occurrence.
L180, L183 respectively to , respectively.
L226 in promoter region to in the promoter region.
L268 was analysed to were analysed.
L272 that were found to were found.
L273 members i.e., to members, i.e.,.
L282, L289, L290, L292, L303 salt tolerant to salt-tolerant.
L301 gene while blue to gene, while blue
L314 ain to response to heat?
L372 role to help to role in help.
L377 superfamily of Trk which to a superfamily of Trk, which.
L402 Additionally, promotes to Additionally, it promotes.
L548 activities i.e., to activities, i.e.,.
I have found plagiarism in this manuscript at L67-68, L152-153, L155-156, L225-226, L249-252, L309, L331-332, L386-388, L389-390, L400-402, L437-440, L451-452, L522-523, L527-528, L528-532, and L538-539. 5% of the article is plagiarized from “Genome-Wide Analysis of Potassium Transport-Related Genes in Chickpea (Cicer arietinum L.) and Their Role in Abiotic Stress Responses” article.
Please clean it.
Reviewer 3 Report
This research work has its merit in the description and classification of the potassium channels of C. cajun, a crop with high economic interest in the region, and its involvement in stress tolerance. The informatic analysis are deep enough for the comprehensive characterization of all the protein sequences but it is lacking in the biological assays since the experimental strategy did not consider some physiological factors regarding stress mechanisms.
Introduction:
What are the concentration ranges found for potassium in the reported plants such as Arabidopsis? Intracellular and extracellular (xylem/phloem).
Are some of the described potassium transporters/channels non-specific for other cations?
What abiotic factors are menacing the C. cajun cultivars in the region? What are the reported environmental changes in the recent decade?
Results:
In Table 1, please indicate in the description the proteins are ordered and grouped according to “Domain” category. Also as the Subcellular localization is Plasma membrane for all, this column can be omitted or only present the Reliability value.
Figure 1 cannot be clearly interpreted since each sequence does not have a label.
Figure 2 should be placed after section 2.3 since this paragraph is completely related with the topic of the figure.
In section 2.2, line 171, there is declared several proteins were not found in the plant’s chromosomes. Is it possible the not found proteins are splicing variants of other genes?
As a curiosity question, can it be determined if the potassium transporters/channels are all coded in the positive strand of the genome or if there are proteins in the negative strand?
In section 2.5, as it was described it gives the impression the experiments were incomplete, since sometimes it talks about salt and drought, the in the next paragraph about salt and heat, etc. It is better to include all the three stress conditions and give the values or tendencies for them all.
In section 2.5.4, since all the antioxidant enzyme activities were decreased under all the stress conditions, it seems C. cajun is a very susceptible species. How can you explain this phenomenon? In other paragraph it was mentioned a tolerant and a susceptible variety of the plant.
In figure 5 I would have liked to see the plant fresh weight and dry weight, since the plant water contents is fully related with the ions concentration and transport as you described in the introduction. And it could have helped for the explanation of the antioxidant enzymes behavior.
In section 2.7, the gene expression changes of the aerial tissue may not be the same as in the root or shoot tissues. Because of this, some of your results of differential gene expression are not in concordance with has been published in other works and do not explain the biological responses.
Discussion:
In the line 386 it was explained salt excess causes damage to protein synthesis, it has more relation with the loss of tertiary structure and then the degradation of misfolded proteins by proteasome.
In the paragraph of the antioxidant activity, the biological response depends on the tissue that is first exposed to the stress factor. Salt and drought first affect the underground tissues, thus the response in roots is different from shoots and from leaves and flowers and seeds. But heat is different since this factor affects first the aerial tissues (leaves, shoots, flowers), while the roots and part of the shoot in protected underground. Because of that the expression patterns should be explained locally and not as systemic response in these cases.
In the line 402, it declares the lipid peroxidation reduces membrane fluidity, please check this fact since the destruction of membrane lipids may increase the fluidity of the membrane and also lead to ion leakage.
Materials and methods:
In section 4.5, How was determined the tolerance limits of the plants for the stress factors not to kill the plants? Maybe the stress factors applied in the experiments were to severe and it was the reason the antioxidant enzymes were reduced instead of increased. How was the plant viability after the experiments? Can you add some photos as supplementary figure?
Were all the plants harvested one week after treatment? The heat stress treatment only says the factor was applied for 8 hours, where the plants immediately harvested after the treatment?
In section 4.8 it declares the RNA was extracted from leaf samples, thus all the results should be explained in terms of the aerial tissue physiological mechanisms for tolerance or susceptibility.
Round 2
Reviewer 2 Report
I am happy with authors comments. Manuscript looks refine and can be accepted in its current format.
Reviewer 3 Report
The new version of the manuscript introduced many of the suggestions and corrected several lines to clarify the described concepts.
All the revision queries have been answered based on bibliografic references and internal laboratory information.
The overall quality of the work has notoriously improved. I recommend the acceptance of this work.